# Parameterized Explainer for Graph Neural Network

**Dongsheng Luo**[1*]   **Wei Cheng**[2*]   **Dongkuan Xu**[1]   **Wenchao Yu**[2]   **Bo Zong**[2]
**Haifeng Chen**[2]   **Xiang Zhang**[1]
[1]The Pennsylvania State University
[2]NEC Labs America
[1]{dul262,dux19,xzz89}@psu.edu
[2]{weicheng,wyu,bzong,haifeng}@nec-labs.com

## Abstract

Despite recent progress in Graph Neural Networks (GNNs), explaining predictions made by GNNs remains a challenging open problem. The leading method independently addresses the local explanations (i.e., important subgraph structure and node features) to interpret why a GNN model makes the prediction for a single instance, e.g. a node or a graph. As a result, the explanation generated is painstakingly customized for each instance. The unique explanation interpreting each instance independently is not sufficient to provide a global understanding of the learned GNN model, leading to the lack of generalizability and hindering it from being used in the inductive setting. Besides, as it is designed for explaining a single instance, it is challenging to explain a set of instances naturally (e.g., graphs of a given class). In this study, we address these key challenges and propose PGExplainer, a parameterized explainer for GNNs. PGExplainer adopts a deep neural network to parameterize the generation process of explanations, which enables PGExplainer a natural approach to explaining multiple instances collectively. Compared to the existing work, PGExplainer has better generalization ability and can be utilized in an inductive setting easily. Experiments on both synthetic and real-life datasets show highly competitive performance with up to 24.7% relative improvement in AUC on explaining graph classification over the leading baseline.

## 1   Introduction

Graph Neural Networks (GNNs) are powerful tools for representation learning of graph-structured data, such as social networks [46], document citation graphs [33], and microbiological graphs [44]. GNNs broadly adopt a message passing scheme to learn node representations by aggregating representation vectors of its neighbors [49, 16]. This scheme enables GNN to capture both node features and graph topology. GNN-based methods have achieved state-of-the-art performance in node classification, graph classification, and link prediction, etc [21, 45, 56].

Despite their remarkable effectiveness, the rationales of predictions made by GNNs are not easy for humans to understand. Since GNNs aggregate both node features and graph topology to make predictions, to understand predictions made by GNNs, important subgraphs and/or a set of features, which are also known as explanations, need to be uncovered. In the literature, although a variety of efforts have been undertaken to interpret general deep neural networks, existing approaches [6, 26, 12, 31, 43, 19, 20] in this line fall short in their ability to explain graph structures, which is essential for GNNs. Explaining predictions made by GNNs remains a challenging open problem, on which few methods have been proposed. The combinatorial nature of explaining graph structures makes it difficult to design models that are both robust and efficient. Recently, the first general model-agnostic

---

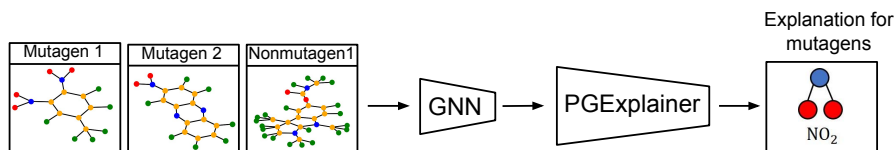

Figure 1: PGExplainer provides human-understandable explanations for predictions made by GNNs. The left part shows the process of applying GNNs for graph classification on the MUTAG dataset. A GNN based model is trained to predict their mutagenic effects. As a post-hoc method, PGExplainer takes the trained GNN model as input and provides consistent explanations for predictions made by the GNN model. For the mutagen molecule graphs in the example, the explanation is the $NO_2$ group.

approach for GNNs, GNNExplainer [53], was proposed to address the problem. It takes a trained GNN and its predictions as inputs to provide interpretable explanations for a given instance, e.g. a node or a graph. The explanation includes a compact subgraph structure and a small subset of node features that are crucial in GNN's prediction for the target instance. Nevertheless, there are several limitations in the existing approach. First, GNNExplainer largely focuses on providing the local interpretability by generating a painstakingly customized explanation for a single instance individually and independently. The explanation provided by GNNExplainer is limited to the single instance, making GNNExplainer difficult to be applied in the inductive setting because the explanations are hard to generalize to other unexplained nodes. As pointed out in previous studies, models interpreting each instance independently are not sufficient to provide a global understanding of the trained model [19]. Furthermore, GNNExplainer has to be retrained for every single explanation. As a result, in real-life scenarios where plenty of nodes need to be interpreted, GNNExplainer would be time-consuming and impractical. Moreover, as GNNExplainer was developed for interpreting individual instances, the explanatory motifs are not learned end-to-end with a global view of the whole GNN model. Thus, it may suffer from suboptimal generalization performance. How to explain predictions of GNNs on a set of instances collectively and easily generalize the learned explainer model to other instances in the inductive setting remains largely unexplored in the literature.

To provide a global understanding of predictions made by GNNs, in this study, we emphasize the *collective* and *inductive* nature of this problem and present our method PGExplainer (Figure 1). PGExplainer is a general explainer that applies to any GNN based models in both transductive and inductive settings. Specifically, a generative probabilistic model for graph data is utilized in PGExplainer. Generative models have shown the power to learn succinct underlying structures from the observed graph data [23]. PGExplainer uncovers these underlying structures as the explanations, which is believed to make the most contribution to GNNs' predictions [35]. We model the underlying structure as edge distributions, where the explanatory graph is sampled. To collectively explain predictions of multiple instances, the generation process in PGExplainer is parameterized with a deep neural network. Since the neural network parameters are shared across the population of explained instances, PGExplainer is naturally applicable to provide model-level explanations for each instance with a global view of the GNN model. Furthermore, PGExplainer has better generalization power because a trained PGExplainer model can be utilized in an inductive setting to infer explanations of unexplained nodes without retraining the explanation model. This also makes PGExplainer much faster than the existing approaches.

Experimental results on both synthetic and real-life datasets demonstrate that PGExplainer can achieve consistent and accurate explanations, bringing up to 24.7% improvement in AUC over the SOTA method on explaining graph classification with significant speed-up.

## 2   Related work

**Graph neural networks**. Graph Neural networks (GNNs) have achieved remarkable success in various tasks, including node classification [25, 21, 45, 48], graph classification [10], and link prediction [56]. The study of GNNs was initiated in [17], and then extended in [41]. These methods iteratively aggregate neighbor information to learn node representations until reaching a static state. Inspired by the success of convolutional neural networks (CNNs) in computer vision, attempts of applying convolutional operations to graphs were derived based on graph spectral theory [4] and

graph Fourier transformation [42]. In recent work, GNNs broadly encode node features as messages and adopt the message passing mechanism to propagate and aggregate them along edges to learn node/graph representations, which are then utilized for downstream tasks [25, 10, 41, 29, 45, 34, 47]. For efficiency consideration, localized filters were proposed to reduce computation cost [21]. The self-attention mechanism was introduced to GNNs in GAT to differentiate the importance of neighbors [45]. Xu. et al. analyzed the relationship between GNNs and Weisfeiler-Lehman graph isomorphism test, and showed the express power of GNNs [49].

**Explaining GNNs.** Interpretability and feature selection have been extensively addressed in neural networks. Methods demystifying complicated deep learning models can be grouped into two main families, whitebox and blackbox [19, 20]. Whitebox mechanisms mainly focus on yielding explanations for individual predictions. Forward and backward propagation based methods are used routinely in whitebox mechanisms. Forward propagation based methods broadly perturb the input and/or hidden representations and check the corresponding updating results in the downstream task [8, 14, 28]. The underlying intuition is that the outputs of the downstream task are likely to significantly change if important features are occluded. Backward propagation based methods, in general, infer important features from the gradients of the deep neural networks. They compute weights of features by propagating the gradients from the output back to the input. Blackbox methods generate explanations by locally learning interpretable models, such as linear models, and additive models to approximate the predictions [32, 5].

Following the line of forward propagation methods, GNNExplainer initiates the research on explaining predictions on graph-structured data [53]. It excludes certain edges and node features to observe the changes in node/graph classification. Explanations (subgraphs/important features) are extracted by maximizing the mutual information between the distribution of possible subgraphs and the GNN's prediction. However, similar to other forward propagation methods, GNNExplainer generates customized explanations for single instance prediction independently, making it insufficient to provide a global understanding of the trained GNN model. Besides, it is naturally difficult to be applied in the inductive setting and provide explanations for multiple instances. XGNN provides model-level explanations without preserving the local fidelity [55]. Thus, the generated explanation may not be a substructure of the real input graph. On the other hand, PGExplainer can provide an explanation for each instance with a global view of the GNN model, which can preserve the local fidelity.

**Graph generation.** PGExplainer learns a probabilistic graph generative model to provide explanations for GNNs. The first model generating random graph is the Erdős–Rényi model [15, 11]. In the random graph proposed by Gilbert, each potential edge is independently chosen from a Bernoulli distribution. Some approaches generate graphs with certain properties reflected, such as pairwise distances betweenness [7], node degree distribution [27], and spectral properties [22, 2]. In recent years, deep learning models have shown great potential to generate graphs with complex properties preserved [54, 18, 35]. However, these methods mainly aim to generate graphs that reflect certain properties in the training graphs.

## 3 Background

We first describe notations, and then provide some background on graph neural networks.

**Notations.** Let $G = (\mathcal{V}, \mathcal{E})$ represent the graph with $\mathcal{V} = \{v_1, v_2 ... v_N\}$ denoting the node set and $\mathcal{E} \in \mathcal{V} \times \mathcal{V}$ as the edge set. The numbers of nodes and edges are denoted by $N$ and $M$, respectively. A graph can be described by an adjacency matrix $\mathbf{A} \in \{0, 1\}^{N \times N}$, with $a_{ij} = 1$ if there is an edge connecting node $i$ and $j$, and $a_{ij} = 0$ otherwise. Nodes in $\mathcal{V}$ are associated with the $d$-dimensional features, denoted by $\mathbf{X} \in \mathbb{R}^{N \times d}$.

**Graph neural networks**. GNNs adopt the message-passing mechanism to propagate and aggregate information along edges in the input graph to learn node representations [25, 21, 45, 16]. Each GNN layer includes three essential steps. First, at the propagation step of the $i$-th GNN layer, for each edge $(i, j)$, GNN computes a message $\mathbf{m}_{ij}^l = \text{Message}(\mathbf{h}_i^{l-1}, \mathbf{h}_j^{l-1})$, where $\mathbf{h}_i^{l-1}$ and $\mathbf{h}_j^{l-1}$ are representations of $v_i$ and $v_j$ in previous layer, respectively. Second, at the aggregation step, for each node $v_i$, GNN aggregates messages received from its neighbor nodes, denoted by $\mathcal{N}_i$, with an aggregation function $\mathbf{m}_{i\cdot}^l = \text{aggregation}(\{\mathbf{m}_{ij}^l | j \in \mathcal{N}_i\})$. Last, at the updating step, GNN updates the vector representation for each node $v_i$ via $\mathbf{h}_i^l = \text{update}(\mathbf{m}_{i\cdot}^l, \mathbf{h}_i^{l-1})$, a function taking the

aggregated message and the representation of itself as inputs. Hidden representation of the last GNN layer serves as the final node representation: $\mathbf{z}_i = \mathbf{h}_i^L$, which is then used for downstream tasks, such as node/graph classification and link prediction.

## 4 The PGExplainer

In this section, we introduce PGExplainer. Different from GNNExplainer which provides explanations on both structure and features, PGExplainer focuses on explanation on graph structures because feature explanation in GNNs is similar to that in non-graph neural networks, which has been extensively studied in the literature [1, 19, 32, 39, 8, 14, 28]. PGExplainer is flexible and applicable to interpret all kinds of GNNs. We start with the learning objective of PGExplainer (Section 4.1) and then present the reparameterization strategy for efficient optimization (Section 4.2). In Section 4.3, we specify particular instantiations to understand GNNs on node and graph classifications. Detailed algorithms can be found in the Appendix.

### 4.1 The learning objective

The literature has shown that real-life graphs are with underling structures [35, 37]. To explain predictions made by a GNN model, we divide the original input graph $G_o$ into two subgraphs: $G_o = G_s + \Delta G$, where $G_s$ presents the underlying subgraph that makes important contributions to GNN's predictions, which is the expected *explanatory graph*, and $\Delta G$ consists of the remaining task-irrelevant edges for predictions made by the GNN. Following [53], PGExplainer finds $G_s$ by maximizing the mutual information between the GNN's predictions and the underlying structure $G_s$:

$$\max_{G_s} \text{MI}(Y_o, G_s) = H(Y_o) - H(Y_o | G = G_s), \tag{1}$$

where $Y_o$ is the prediction of the GNN model with $G_o$ as the input. The mutual information quantifies the probability of prediction $Y_o$ when the input graph to the GNN model is limited to the explanatory graph $G_s$. The intuition behind comes from the traditional forward propagation based methods for the whitebox explanation [8]. For example, if removing an edge $(i, j)$ dramatically changes the prediction in the GNN, then this edge is important and should be included in $G_s$. Otherwise, it can be considered as irrelevant edge for the GNN model to make the prediction. Since $H(Y_o)$ is only related to the GNN model whose parameters are fixed in the explanation stage, the objective is equivalent to minimizing the conditional entropy $H(Y_o | G = G_s)$.

However, the direct optimization of the above objective function is intractable as there are $2^M$ candidates for $G_s$. Thus, we consider a relaxation by assuming that the explanatory graph is a Gilbert random graph [15], where selections of edges from the original input graph $G_o$ are conditionally independent to each other. Let $e_{ij} \in \mathcal{V} \times \mathcal{V}$ be the binary variable indicating whether the edge is selected, with $e_{ij} = 1$ if the edge $(i, j)$ is selected, and 0 otherwise. Let $G$ be the random graph variable. Based on the above assumption, the probability of a graph $G$ can be factorized as:

$$P(G) = \Pi_{(i,j) \in \mathcal{E}} P(e_{ij}). \tag{2}$$

A straightforward instantiation of $P(e_{ij})$ is the Bernoulli distribution $e_{ij} \sim Bern(\theta_{ij})$. $P(e_{ij} = 1) = \theta_{ij}$ is the probability that edge $(i, j)$ exists in $G$. With this relaxation, we thus can rewrite the objective as:

$$\min_{G_s} H(Y_o | G = G_s) = \min_{G_s} \mathbb{E}_{G_s}[H(Y_o | G = G_s)] \approx \min_{\Theta} \mathbb{E}_{G_s \sim q(\Theta)}[H(Y_o | G = G_s)] \tag{3}$$

where $q(\Theta)$ is the distribution of the explanatory graph parameterized by $\theta$'s.

### 4.2 The reparameterization trick

Due to the discrete nature of $G_s$, we relax edge weights from binary variables to continuous variables in the range $(0, 1)$ and adopt the reparameterization trick to efficiently optimize the objective function with gradient-based methods [24]. We approximate the sampling process $G_s \sim q(\Theta)$ with a determinant function of parameters $\Omega$, temperature $\tau$, and an independent random variable $\epsilon$: $G_s \approx \hat{G}_s = f_\Omega(G_o, \tau, \epsilon)$. The temperature $\tau$ is used to control the approximation. Here we utilize the

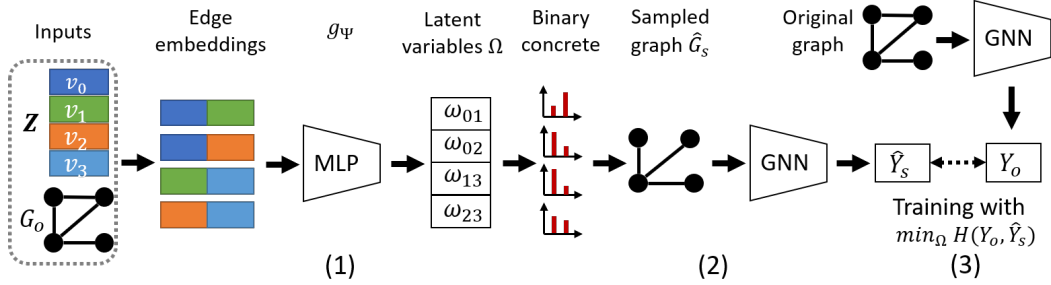

Figure 2: Illustration of PGExplainer for explaining GNNs on graph classification. (1) The left part demonstrates the explanation network. It takes node representations $\mathbf{Z}$ as well as the original graph $G_o$ as inputs to compute $\Omega$, the latent variables in edge distributions. Edge distributions are severed as the explanation. In case that an explanatory subgraph is wanted, we select top-ranked edges according to latent variables $\Omega$. (2) A random graph $\hat{G}_s$ is sampled from edge distributions and then feed to the trained GNN model to get the prediction $\hat{Y}_s$. (3) Parameter $\Psi$ in the explanation network is optimized with cross-entropy between the original prediction $Y_o$ and the updated prediction $\hat{Y}_s$.

binary concrete distribution as the instantiation [36]. Specifically, the weight $\hat{e}_{ij} \in (0, 1)$ of edge $(i, j)$ in $\hat{G}_s$ is calculated by:

$$\epsilon \sim \text{Uniform}(0, 1), \quad \hat{e}_{ij} = \sigma((\log \epsilon - \log(1 - \epsilon) + \omega_{ij})/\tau), \tag{4}$$

where $\sigma(\cdot)$ is the Sigmoid function, and $\omega_{ij} \in \mathbb{R}$ is the parameter. When $\tau \to 0$, the weight $\hat{e}_{ij}$ is binarized with $\lim_{\tau \to 0} P(\hat{e}_{ij} = 1) = \frac{\exp(\omega_{ij})}{1 + \exp(\omega_{ij})}$. Since $P(e_{ij} = 1) = \theta_{ij}$, by choosing $\omega_{ij} = \log \frac{\theta_{ij}}{1 - \theta_{ij}}$, we have $\lim_{\tau \to 0} \hat{G}_s = G_s$. This demonstrates the rationality of using binary concrete distribution to approximate the Bernoulli distribution. Moreover, with temperature $\tau > 0$, the objective function is smoothed with a well-defined gradient $\frac{\partial \hat{e}_{ij}}{\partial \omega_{ij}}$. Thus, with reparameterization, the objective in Eq. (3) becomes:

$$\min_{\Omega} \mathbb{E}_{\epsilon \sim \text{Uniform}(0,1)} H(Y_o | G = \hat{G}_s) \tag{5}$$

Considering efficient optimization, we follow [53] to modify the conditional entropy with cross-entropy $H(Y_o, \hat{Y}_s)$, where $\hat{Y}_s$ is the prediction of the GNN model with $\hat{G}_s$ as the input. With the above relaxations, we adopt Monte Carlo to approximately optimize the objective function:

$$\min_{\Omega} \mathbb{E}_{\epsilon \sim \text{Uniform}(0,1)} H(Y_o, \hat{Y}_s) \approx \min_{\Omega} -\frac{1}{K} \sum_{k=1}^{K} \sum_{c=1}^{C} P(Y_o = c) \log P(\hat{Y}_s = c)$$

$$= \min_{\Omega} -\frac{1}{K} \sum_{k=1}^{K} \sum_{c=1}^{C} P_\Phi(Y = c | G = G_o) \log P_\Phi(Y = c | G = \hat{G}_s^{(k)}), \tag{6}$$

where $\Phi$ denotes the parameters in the trained GNN, $K$ is the total number of sampled graph, $C$ is the number of labels, and $\hat{G}_s^{(k)}$ is the $k$-th sampled graph with Eq. (4), parameterized by $\Omega$.

## 4.3 Explanation of graph neural networks with a global view

Although explanations provided by the leading method GNNExplainer [53] preserve the local fidelity, they do not help to understand the general picture of the model across a population [50]. Furthermore, various GNN based models have been applied to analyze graph data with millions of instances [52], the cost of applying local explanations one-by-one can be prohibitive with such large datasets in practice. On the other hand, explanations with a global view of the model ascertain users' trust [39]. Furthermore, these models can generalize explanations to new instances without retraining, making it more efficient to explain large scale datasets.

To have a global view of a GNN model, our method collectively explains predictions made by a trained model on multiple instances. Instead of treating $\Omega$ in Eq. (6) as independent variables, we

utilize a parameterized network to learn to generate explanations from the trained GNN model, which also applies to unexplained instances. In general, GNN based models first learn node representations and then feed the vector representations to downstream tasks [21, 25, 45]. We denote these two functions by $\text{GNNE}_{\Phi_0}(\cdot)$ and $\text{GNNC}_{\Phi_1}(\cdot)$, respectively. For GNNs without explicit classification layers, we use the last layer instead. As a result, we can represent a GNN model with:

$$\mathbf{Z} = \text{GNNE}_{\Phi_0}(G_o, \mathbf{X}), \ \ Y = \text{GNNC}_{\Phi_1}(\mathbf{Z}). \tag{7}$$

$\mathbf{Z}$ is the matrix of node representations encoding both features and structure of the input graph, which is used as an input in the explanation network to calculate the parameter $\Omega$:

$$\Omega = g_\Psi(G_o, \mathbf{Z}). \tag{8}$$

$\Psi$ denotes parameters in the explanation network, which is shared by all edges among the population. Therefore, PGExplainer can be utilized to collectively provide explanations for multiple instances. Specifically, in the collective setting with instance set $\mathcal{I}$, the objective of PGExplainer is:

$$\min_\Psi -\sum_{i \in \mathcal{I}} \sum_{k=1}^{K} \sum_{c=1}^{C} P_\Phi(Y = c | G = G_o^{(i)}) \log P_\Phi(Y = c | G = \hat{G}_s^{(i,k)}), \tag{9}$$

where $G^{(i)}$ is the input graph and $\hat{G}_s^{(i,k)}$ is the $k$-th sampled graph with Eq. (4 ,8) for $i$-th instance. We consider both graph and node classifications and specify an instantiation for each task. Solutions for other tasks, such as link prediction, are similar and thus omitted.

**Explanation network for node classification.** Considering that explanations for nodes in a graph may appear diverse structures [53], especially for nodes with different labels. For example, an edge $(i, j)$ is important for the prediction of node $u$, but not for another node $v$. Based on this motivation, we implement the $\Omega = g_\Psi(G_o, \mathbf{Z})$ to explain the prediction of node $v$ with:

$$\omega_{ij} = \text{MLP}_\Psi([\mathbf{z}_i; \mathbf{z}_j; \mathbf{z}_v]). \tag{10}$$

$\text{MLP}_\Psi$ is a multi-layer neural network parameterized with $\Psi$ and $[\cdot; \cdot]$ is the concatenation operation.

**Explanation network for graph classification**. For graph level tasks, each graph is considered as an instance. The explanation of the prediction of a graph is not conditional to a specific node. Therefore, we specify the $\Omega = g_\Psi(G_o, \mathbf{Z})$ for graph classification as:

$$\omega_{ij} = \text{MLP}_\Psi([\mathbf{z}_i; \mathbf{z}_j]). \tag{11}$$

With the graph classification as an example, the architecture of PGExplainer is shown in Figure 2. The algorithms of PGExplainer for node and graph classification can be found in Appendix.

**Computational complexity**. PGExplainer is more efficient than GNNExplainer for two reasons. First, PGExplainer learns a latent variable for each edge in the original input graph with a neural network parameterized by $\Psi$, which is shared by all edges in the population of input graphs. Different from GNNExplainer, whose parameter size is linear to the number of edges, the number of parameters in PGExplainer is irrelevant to the size of the input graph, which makes PGExplainer applicable to large scale datasets. Further, since the explanation is shared among a population, a trained PGExplainer can be utilized in the inductive setting to explain new instances without retraining the explainer. To explain a new instance with $|\mathcal{E}|$ edges in the input graph, the time complexity of PGExplainer is $O(|\mathcal{E}|)$. As a comparison, GNNExplainer has to retrain for the new instance, leading to the time complexity of $O(T|\mathcal{E}|)$, where $T$ is the number of epochs for retraining.

## 4.4 Regularization

The framework of PGExplainer is flexible with various regularization terms to preserve desired properties on the explanation. We now discuss the regularization terms as well as their principles.

**Size and entropy constraints.** Following [53], to obtain compact and succinct explanations, we impose a constraint on the explanation size by adding $||\Omega||_1$, the $l_1$ norm on latent variables $\Omega$, as a regularization term. Besides, element-wise entropy is also be applied to further achieve discrete edge weights [53].

Next, We provide more regularization terms that are compatible with PGExplainer. Note that for a fair comparison, the following regularization terms are not utilized in experimental studies in Section 5.

**Budget constraint.** To obtain a compact explanation, the $l_1$ norm on latent variables $\Omega$ is introduced, which penalizes all edge weights to sparsify the explanatory graph. In cases that a predefined budget $B$ is available, for example, $|G_s| \leq B$, we could modify the size constraint to budget constraint:

$$R_b = \text{ReLU}( \sum_{(i,j) \in \mathcal{E}} \hat{e}_{ij} - B). \tag{12}$$

When the size of the explanatory graph is smaller than the budget $B$, the budget regularization $R_b = 0$. On the other hand, it works similarly to the size constraint when out of budget.

**Connectivity constraint.** In many real-life scenarios, determinant motifs are expected to be connected. Although it is claimed that GNNExplainer empirically tends to detect a small connected subgraph, the explicit constraints are not provided [53]. We propose to implement the connected constraint with the cross-entropy of adjacent edges, which connect to the same node. For instance, $(i,j)$ and $(i,k)$ both connected to the node $i$. The motivation is that is $(i,j)$ is selected in the the explanatory graph, then its adjacent edge $(i,k)$ should also be included. Formally, we design the connectivity constraint as:

$$H(\hat{e}_{ij}, \hat{e}_{ik}) = -[(1 - \hat{e}_{ij}) \log(1 - \hat{e}_{ik}) + \hat{e}_{ij} \log \hat{e}_{ik}]. \tag{13}$$

## 5 Experimental study

In this section, we evaluate our PGExplainer with a number of experiments. We first describe synthetic and real-world datasets, baseline methods, and experimental setup. Then, we present the experimental results on explanations of both node and graph classification. With qualitative and quantitative evaluations, we demonstrate that our PGExplainer can improve the SOTA method up to 24.7% in AUC on explaining graph classification. At the same time, with a trained explanation network, our PGExplainer is significantly faster than the baseline when explaining unexplained instances. We also provide extended experiments to show deep insights of PGExplainer in the appendix. The code and data used in this work are available [2].

### 5.1 Datasets

We follow the setting in GNNExplainer and construct four kinds of node classification datasets, BA-Shapes, BA-Community, Tree-Cycles, and Tree-Grids [53]. Furthermore, we also construct a graph classification datasets, BA-2motifs. Illustration of synthetic datasets is shown in Table 2. (1) BA-Shapes is a single graph consisting of a base Barabasi-Albert (BA) graph with 300 nodes and 80 "house"-structured motifs. These motifs are attached to randomly selected nodes from the BA graph. After that, random edges are added to perturb the graph. Nodes features are not assigned in BA-Shapes. Nodes in the base graph are labeled with 0; the ones locating at the top/middle/bottom of the "house" are labeled with 1,2,3, respectively. (2) BA-Community dataset consists of two BA-Shapes graphs. Two Gaussian distributions are utilized to sample node features, one for each BA-Shapes graph. Nodes are labeled based on their structural roles and community memberships, leading to 8 classes in total. (3) In the Tree-Cycles dataset, an 8-level balanced binary tree is adopted as the base graph. A set of 80 six-node cycle motifs are attached to randomly selected nodes from the base graph. (4) Tree-Grid is constructed in the same way as TREE-CYCLES, except that 3-by-3 grid motifs are used to replace the cycle motifs. (5) For graph classification, we build the BA-2motifs dataset of 800 graphs. We adopt the BA graphs as base graphs. Half graphs are attached with "house" motifs and the rest are attached with five-node cycle motifs. Graphs are assigned to one of 2 classes according to the type of attached motifs.

We also include a real-life dataset, MUTAG, for graph classification, which is also used in previous work [53]. It consists of 4,337 molecule graphs. Each graph is assigned to one of 2 classes based on its mutagenic effect [53, 40]. As discussed in [53, 9], carbon rings with chemical groups $NH_2$ or $NO_2$

are known to be mutagenic. We observe that carbon rings exist in both mutagen and nonmutagenic graphs, which are not discriminative. Thus, we can treat carbon rings as the shared base graphs and $NH_2$, $NO_2$ as motifs for the mutagen graphs. There are no explicit motifs for nonmutagenic ones. Table 1 shows the statistics of synthetic and real-life datasets.

Table 1: Dataset statistics

|  | Node Classification | | | | Graph Classification | |
|---|---|---|---|---|---|---|
|  | BA-Shapes | BA-Community | Tree-Cycles | Tree-Grid | BA-2motifs | MUTAG |
| #graphs | 1 | 1 | 1 | 1 | 1,000 | 4,337 |
| #nodes | 700 | 1,400 | 871 | 1,231 | 25,000 | 131,488 |
| #edges | 4,110 | 8,920 | 1,950 | 3,410 | 51,392 | 266,894 |
| #labels | 4 | 8 | 2 | 2 | 2 | 2 |

## 5.2 Baselines and experimental setup

**Baselines.** We compare with the following baseline methods, GNNExplainer [53], a gradient-based method (GRAD) [53], graph attention network (ATT) [45], and Gradient [38]. (1) GNNExplainer is a post-hoc method providing explanations for every single instance. (2) GRAD learns weights of edges by computing gradients of GNN's objective function w.r.t. the adjacency matrix. (3) ATT utilizes self-attention layers to distinguish edge attention weights in the input graph. Each edge's importance is obtained by averaging its attention weights across all attention layers. (4) We first adopt Gradient in [38] to calculate the importance of each node, then calculate the importance of an edge by average the connected nodes' importance scores.

**Experimental setup.** We follow the experimental settings in GNNExplainer [53]. Specifically, for post-hoc methods including ATT, GNNExplainer, and PGExplainer, we first train a three-layer GNN and then apply these methods to explain predictions made by the GNN. Since weights in attention layers are jointly optimized with the GNN model in ATT, we thus train another GNN model with self-attention layers. We follow [1] to tune temperature $\tau$. We refer readers to the Appendix for more training details.

## 5.3 Results

The results of comparative evaluation experiments on both synthetic and real-life datasets are summarized in Table 2. In these datasets, node/graph labels are determined by the motifs, which are treated as ground truth explanations. These motifs are utilized to calculate explanation accuracy for PGExplainer as well as other baselines.

**Qualitative evaluation.** We choose an instance for each dataset and visualize its explanations given by GNNExplainer and PGExplainer in Table 2. In these explanations, bold black edges indicate top-$K$ edges ranked by their importance weights, where $K$ is set to the number of edges inside motifs for synthetic datasets and 10 for MUTAG [53]. As demonstrated in these figures, the whole motifs, such as "house" in BA-Shapes and BA-Community, cycles in Tree-Cycles and BA-2motifs, grids in Tree-Grid, and $NO_2$ groups in MUTAG are correctly identified by PGExplainer. On the other hand, some important edges are missing in the explanations given by GNNExplainer. For example, the explanation provided by GNNExplainer for the instance in MUTAG contains the carbon rings and part of a $NO_2$ group. However, the carbon rings appear frequently in both mutagen and nonmutagenic graphs, which are not discriminative. Conversely, PGExplainer correctly identifies both $NO_2$ groups.

**Quantitative evaluation.** We follow the experimental settings in GNNExplainer [53] and formalize the explanation problem as a binary classification of edges. We treat edges inside motifs as positive edges, and negative otherwise. Importance weights provided by explanation methods are considered as prediction scores. A good explanation method assigns high weights to edges in the ground truth motifs than the ones outside. AUC is adopted as the metric for quantitative evaluation. Especially, for the MUTAG dataset, we only consider the mutagen graphs because no explicit motifs exist in nonmutagenic ones. For PGExplainer, we repeat each experiment 10 times and report the average AUC scores and standard deviations here.

From the table, we have the following observations. PGExplainer achieves SOTA performances in all scenarios and the accuracy gains are up to 13.0% in node classification and 24.7% in graph

Table 2: Illustration of different datasets together with performance evaluation of PGExplainer and other baselines. BA-Shapes, BA-Community, Tree-Cycles, Tree-Grid are datasets for node classification [53]. Node labels are represented by their colors. BA-2motifs and MUTAG datasets are used for graph classification. Graphs with "house" motifs are labeled with 0 and the ones with cycles are with 1 in BA-2motifs dataset. $NH_2$, $NO_2$ are treated as motifs of the mutagen graphs in MUTAG. Explanations extracted by GNNExplainer and PGExplainer are also shown as case studies.

**Explanation AUC**

|  | BA-Shapes | BA-Community | Tree-Cycles | Tree-Grid | BA-2motifs | MUTAG |
|---|---|---|---|---|---|---|
| GRAD | 0.882 | 0.750 | 0.905 | 0.612 | 0.717 | 0.783 |
| ATT | 0.815 | 0.739 | 0.824 | 0.667 | 0.674 | 0.765 |
| Gradient | - | - | - | - | 0.773 | 0.653 |
| GNNExplainer | 0.925 | 0.836 | 0.948 | 0.875 | 0.742 | 0.727 |
| PGExplainer | **0.963**±0.011 | **0.945**±0.019 | **0.987**±0.007 | **0.907**±0.014 | **0.926**±0.021 | **0.873**±0.013 |
| Improve | 4.1% | 13.0% | 4.1% | 3.7% | 24.7% | 11.5% |

**Inference Time (ms)**

|  | BA-Shapes | BA-Community | Tree-Cycles | Tree-Grid | BA-2motifs | MUTAG |
|---|---|---|---|---|---|---|
| GNNExplainer | 650.60 | 696.61 | 690.13 | 713.40 | 934.72 | 409.98 |
| PGExplainer | 10.92 | 24.07 | 6.36 | 6.72 | 80.13 | 9.68 |
| Speed-up | 59x | 29x | 108x | 106x | 12x | 42x |

classification. Compared to GNNExplainer, which tackles instances independently thus can only achieve suboptimal explanations, PGExplainer utilizes a parameterized explanation network based upon graph generative model to collectively provide explanations for multiple instances. As a result, PGExplainer can have a global view of the GNNs, which answers why PGExplainer can outperform GNNExplainer by relatively large margins.

**Efficiency evaluation.** Explanation network in PGExplainer is shared across the population of instances. Thus, a trained PGExplainer can be utilized to explain new instances in the inductive setting. We denote the time to explain a new instance with a trained explanation method by inference time. Since GNNExplainer has to retrain the model, we also count the training time here. The running time comparison in Table 2 shows that PGExplainer can speed up the computation significantly, up to 108 times faster than GNNExplainer, which makes PGExplainer more practical for large-scale datasets.

Further experiments on the inductive performance and effects of regularization terms are in Appendix.

## 6   Conclusion

We present PGExplainer, a parameterized method to provide a global understanding of any GNN models on arbitrary machine learning tasks by collectively explaining multiple instances. We show that PGExplainer can leverage the representations produced by GNNs to learn the underlying subgraphs that are important to the predictions made by GNNs. Furthermore, PGExplainer is more efficient due to its capacity to explain GNNs in the inductive settings, which makes PGExplainer more practical in real-life applications.

## Acknowledgement

This project was partially supported by NSF projects IIS-1707548 and CBET-1638320.

## Broader impact

Graph neural networks are powerful tools that have been applied in various real-world applications, including community detection, recommendation systems, computer vision, and natural language processing [3, 13, 30, 51]. Our work can not only provide interpretable explanations with local fidelity for predictions made by GNN models, but also improve the global understanding of the model.

There are several broader impacts of using our method to explain predictions made by GNNs. First, our method can increase the transparency of applying GNNs for decision-critical applications, such as drug discovery and diagnosis. As a result, our method can help alleviate safety, and fairness risks. For example, as we show in our experiments, we could correctly identify motifs that have determinant effects on the mutagenicity of molecules. On the other hand, our method also puts GNN models at a high risk of being attacked. Our method extracts subgraphs that are important to GNNs' behaviors. Disturbing these parts leads to significant changes in GNNs' predictions. Besides, increasing the interpretability of GNNs may cause automation bias, such as an undue trust on GNN models.

## Footnotes

[2]`https://github.com/flyingdoog/PGExplainer`

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
