[Supplementary Material]

# Supplementary Material: Parameterized Explainer for Graph Neural Network

## A. Explanation algorithms

The algorithms of PGExplainer for node and graph classification are shown in Algorithm 1 and 2, respectively. We first discuss the node classification in Algorithm 1. In GNNs with message passing

---

**Algorithm 1:** Training algorithm for explaining node classification

---

1: **Input:** The input graph $G_o = (\mathcal{V}, \mathcal{E})$, node features $\mathbf{X}$, node labels $Y$, the set of instances to be explained $\mathcal{I}$, a trained GNN model: $\text{GNNE}_{\Phi_0}(\cdot)$ and $\text{GNNC}_{\Phi_1}(\cdot)$.
2: **for** each node $i \in \mathcal{I}$ **do**
3:      $G_o^{(i)} \leftarrow$ extract the computation graph for node $i$.
4:      $\mathbf{Z}^{(i)} \leftarrow \text{GNNE}_{\Phi_0}(G_o^{(i)}, \mathbf{X})$.
5:      $Y_o^{(i)} \leftarrow \text{GNNC}_{\Phi_1}(\mathbf{Z}^{(i)})$.
6: **end for**
7: **for** each epoch **do**
8:      **for** each node $i \in \mathcal{I}$ **do**
9:          $\Omega \leftarrow$ latent variables calculated with Eq. (10)
10:          **for** $k \leftarrow 1$ **to** $K$ **do**
11:              $\hat{G}_s^{(i,k)} \leftarrow$ sampled from Eq. (4).
12:              $\hat{Y}_s^{(i,k)} \leftarrow \text{GNNC}_{\Phi_1}(\text{GNNE}_{\Phi_0}(\hat{G}_s^{(i,k)}, \mathbf{X}))$
13:          **end for**
14:      **end for**
15:      Compute loss with Eq. (9).
16:      Update parameters $\Psi$ with backpropagation.
17: **end for**

---

mechanisms, the prediction at a node $v$ is fully determined by its local computation graph, which is defined by its $L$-hop neighborhoods [53]. $L$ is the number of GNN layers. Thus, for each node $i$ in the instance set $\mathcal{I}$ to be explained, we first extract a local computation graph $G_o^{(i)}$ (line 3). With $G_o^{(i)}$ as the input graph, the trained GNN model generates the label of node $i$, denoted by $Y_o^{(i)}$ (line 4-5). To train the explanation network, each time we select a node $i$ and compute parameters $\Omega$ in edge distributions with Eq. (10) (line 9). After that, we sample $K$ graphs as input graphs for GNN to get updated predictions for node $i$, with the $k$-th prediction denoted by $\hat{Y}_s^{(i,k)}$ (line 11-13). We compute the loss and update parameters $\Psi$ in the explanation network in line 15-16.

---

**Algorithm 2:** Training algorithm for explaining graph classification

---

1: **Input:** A set of input graphs with $i$-th graph represented by $G_o^{(i)}$, node features $\mathbf{X}^{(i)}$, and a label $Y^{(i)}$, a trained GNN model: $\text{GNNE}_{\Phi_0}(\cdot)$ and $\text{GNNC}_{\Phi_1}(\cdot)$.
2: **for** each graph $G_o^{(i)}$ **do**
3:      $\mathbf{Z}^{(i)} \leftarrow \text{GNNE}_{\Phi_0}(G_o^{(i)}, \mathbf{X}^{(i)})$.
4:      $Y_o^{(i)} \leftarrow \text{GNNC}_{\Phi_1}(\mathbf{Z}^{(i)})$.
5: **end for**
6: **for** each epoch **do**
7:      **for** each graph $G_o^{(i)}$ **do**
8:          $\Omega \leftarrow$ latent variables calculated with Eq. (11)
9:          **for** $k \leftarrow 1$ **to** $K$ **do**
10:              $\hat{G}_s^{(i,k)} \leftarrow$ sampled from Eq. (4).
11:              $\hat{Y}_s^{(i,k)} \leftarrow \text{GNNC}_{\Phi_1}(\text{GNNE}_{\Phi_0}(\hat{G}_s^{(i,k)}, \mathbf{X}^{(i)}))$
12:          **end for**
13:      **end for**
14:      Compute loss with Eq. (9).
15:      Update parameters $\Psi$ with backpropagation.
16: **end for**

---

The training algorithm for explaining graph classification is shown in Algorithm 2. The algorithm is similar to the one explaining node classifications, except that computation graphs are not used, because, for graph classification, each graph is treated as an instance. Given a set of graphs $\{G_o^{(i)}\}_{i \in \mathcal{I}}$, we first compute the node embeddings $\mathbf{Z}^{(i)}$ and graph labels $Y_o^{(i)}$ with the trained GNN model (line 2-4). In each epoch, for each $i$-th graph, we compute the parameters $\Omega$ in its edge distributions with Eq. 11 (line 8). We then sample $K$ subgraphs and get the updated predictions. We compute the loss with Eq. (9) and update parameters $\Psi$ with backpropagation.

## B. Hardware and implementations in experiments

All experiments are conducted on a Linux machine with an Nvidia GeForce RTX 2070 SUPER GPU with 8GB memory. CUDA version is 10.2 and Driver Version is 440.64.00. PGExplainer is implemented with Tensorflow 2.0.0. For each dataset, we first train a GNN model, which is then shared by all posthoc methods ATT, GNNExplainer, and PGExplainer. We use $FC(a, b, f)$ to denote a fully-connected layer. $a$ and $b$ are the numbers of input and output neurons respectively. $f$ is the activation function. Similarly, we denote a GNN layer with input dimension $a$, output dimension $b$, and activation function $f$ by $GNN(a, b, f)$. With these notations, the network structure of the GNN model for node classification is GNN(10, 20, ReLU)-GNN(20, 20, ReLU)-GNN(20, 20, ReLU)-FC(20, #label, softmax). For graph classification, we add a maxpooling layer to get graph representations before the final FC layer. Thus, the network structure is GNN(10, 20, ReLU)-GNN(20, 20, ReLU)-GNN(20, 20, ReLU)-Maxpooling-FC(20, #label, softmax). We adopt the Adam optimizer with the initial learning rate of $1.0 \times 10^{-3}$. All variables are initialized with Xavier. We follow GNNExplainer to split train/validation/test with 80/10/10% for all datasets. Each model is trained for 1000 epochs. The accuracy performances of GNN models are shown in Table 3. The results show that the designed GNN models are powerful enough for node/graph classifications on both synthetic and real-life datasets.

Table 3: Accuracy performance of GNN models

| Accuracy | Node Classification | | | | Graph Classification | |
| --- | --- | --- | --- | --- | --- | --- |
| | BA-Shapes | BA-Community | Tree-Cycles | Tree-Grid | BA-2motifs | MUTAG |
| Training | 0.98 | 0.99 | 0.99 | 0.92 | 1.00 | 0.87 |
| Validation | 1.00 | 0.88 | 1.00 | 0.94 | 1.00 | 0.89 |
| Testing | 0.97 | 0.93 | 0.99 | 0.94 | 1.00 | 0.87 |

The network structure of explanation networks in PGExplainer is FC(#input, 64, ReLU)-FC(20, 1, Linear), which is shared for all datasets. #input is 60 for node classification, and 40 for graph classification.To train PGExplainer, we also adopt the Adam optimizer with the initial learning rate of $3.0 \times 10^{-3}$. The coefficient of size regularization is set to 0.05 and entropy regularization is 1.0. The epoch $T$ is set to 30 for all datasets. The temperature $\tau$ in Eq. (4) is set with annealing schedule [1]: $\tau^{(t)} = \tau_0(\tau_T/\tau_0)^t$, where $\tau_0$ and $\tau_T$ are the initial and final temperatures. A small temperature tends to generate more discrete graphs which may hinder the explanation network being optimized with backpropagation. In this task, we find that relatively high temperatures work well in practice. $\tau_0$ is set to 5.0 and $\tau_T$ is set to 2.0.

## C. Additional experiments

In this part, we conduct extensive experiments to have deep insights into our PGExplainer.

### C.1 Inductive performance

As we discussed in Section 4.3, the explanation network is shared across the population. Thus, with a trained PGExplainer, we can directly infer the explanation without retraining the explanation network. As a result, our PGExplainer has better generalization power than the leading baseline GNNExplainer. Besides, our PGExplainer is more efficient in the inductive setting. In this section, we empirically demonstrate the effectiveness of PGExplainer in the inductive setting. In the inductive setting, we select $\alpha$ instances for training, $(N - \alpha)/2$ for validation, and the rest for testing. $\alpha$ is ranged from $[1, 2, 3, 4, 5, 30]$. Note that, with $\alpha = 1$, our method degenerates to the single-instance explanation

method. Recall that to explain a set of instances, GNNExplainer first detects a reference node and then computes the explanation for the reference node. The explanation is then generalized to other nodes with graph alignment [53]. We claim that it may lead to sub-optimal explanations because reference node selection and graph alignment are not jointly optimized with the explanation in an end-to-end fashion. The AUC scores of PGExplainer are shown in Figure 3. We have the following observations. 1) The testing AUC increase as more instances are trained, verifying the effectiveness of PGExplainer. Some results are higher than the reported ones in Section 5 because here we adopt validation datasets to fine-tune the hyper-parameters. 2) More training instances lead to smaller standard deviation and PGExplainer tends to globally detect shared motifs with higher robustness. 3) PGExplainer can achieve relatively good performance with a small number of trained instances, which makes PGExplainer more practical in large datasets. The results also explain why we dismiss the training time of PGExplainer and only count the inference time in Section 5.

(a) BA-Shapes  (b) BA-Community  (c) Tree-Cycles

(d) Tree-Grid  (e) BA-2motifs

Figure 3: Evaluation of PGExplainer in the inductive setting.

## C.2 Effects of regularization terms

In this part, we analyze the effects of regularization terms. In addition to the size and entropy regularizers introduced in GNNExplainer, we also have discussed regularization terms on budgets and connectivity constraints. Since the first two regularizers are used in the quantitative evaluation, we first conduct parameter studies. Visualization results on synthetic datasets show that the explanatory graph extract by PGExplainer tends to be small and compact. To verify the effectiveness of the proposed regularizer for connectivity constraint, we synthesize a noisy BA-Shapes dataset.

**Effects of size and entropy constraint.** We select synthetic datasets for parameter study. The coefficients of size and entropy regularizers are denoted by $\lambda_s$ and $\lambda_e$, respectively. AUC scores w.r.t coefficients are shown in Figure 4. We observe that PGExplainer achieves competitive performances even without any regularization terms in all datasets except the BA-Community, which verifies the effectiveness of the model itself. For the BA-Community dataset, the entropy constraint plays an important role.

**Effects of connectivity constraint.** To show the effect of the connectivity constraint on the explanatory graph, we build a noisy BA-Shapes dataset with $0.2N$ noisy edges. We vary the coefficient of the connectivity regularization term $\lambda_c$ from 0 to 10 and apply PGExplainer to explain a single instance.

(a) BA-Shapes       (b) BA-Community       (c) Tree-Cycles

(d) Tree-Grid       (e) BA-2motifs

Figure 4: Effects of size and entropy constraint

The visualization results with regard to different choices of coefficients are shown in Figure 5. The figure demonstrates that without explicit constraint, PGExplainer may detect several connected edges in the noisy input graph, although these edges are also inside motifs. With the connectivity constraint, we observe that PGExplainer tends to provide a connected subgraph as an explanation.

(a) $\lambda_c = 0$       (b) $\lambda_c = 5$       (c) $\lambda_c = 10$

Figure 5: Effects of connectivity constraint

## D. Selection of subset of features

In this paper, we focus on globally understanding predictions made by GNNs by providing topological explanations. To explain node features, in GNNExplainer, the authors propose to use a feature mask to select features that are important to preserve original predictions. Feature selection has been extensively studied in non-graph neural networks and can be applied directly to explain GNNs, such as the concrete autoencoder [1]. Besides, since the selected features are shared among instances across the population, feature explanation is naturally global and applicable to new instances in the inductive setting.