[Reviews · NeurIPS 2020]

Review 1

Summary and Contributions: The paper discusses a post-hoc explainer for graph neural networks which leverages a generative approach that samples Gilbert random graphs as candidate explanatory substructures. The explainer is trained as in GNNExplainer to optimise the substructure-conditioned entropy of the predicted class. The proposed approach can be used in multi-instance settings and it is empirically validated against the models discussed in the original GNNExplainer paper (in similar experimental setting).

Strengths: The topic of explainability in graph neural networks is certainly of high interest for the community and it has been, yet, poorly explored. As such the work as very high potential impact. The proposed approach tackles specifically the issue of providing a model-level (or multi-instance) explanation, which is even less covered by the works in literature (and hence more novel). However, the Authors should note that it is not true that GNN-explainer cannot work in multi-instance settings (more on this later). POST REBUTTAL: After reading the rebuttal and the discussion with the reviewers, it is clear that the paper generates some confusion on the nature of the explanations. While the Authors claim the explanations to be somewhat "global", it seems that what is being provided is more of a model explanation generated locally. I think this confusion needs to be clearly addressed by the Authors, and it is also the result of a poor positioning of the paper w.r.t. the literature. So I warmly invite the Authors to reconsider carefully the placement of their proposed model with respect to the interpretability taxonomy. Novelty-wise, the work reuses many known concepts and methodologies (such as Gilbert random graph sampling from soft-binomials and conditional-entropy optimization as explanation objective) but combines them in a sufficiently original way and the resulting model appears effective (although some caution should be used on the empirical results until some open questions are answered). POST REBUTTAL: After the discussion it seems that the proposed model is somewhat original, as it builds on ideas presented already elsewhere. But this is the first time they are implemented in the context of interpretability on graphs. This makes the paper sort of borderline.

Weaknesses: The work has a certain tendency to misrepresent the literature for the sake of highlighting the originality and impact of this paper. I do not particularly appreciate this type of approach to a scientific paper. More specifically, I strongly invite the Authors to reconsider the following aspects: - Multi-instance Vs Single-Instance: much of the introduction and motivation of the paper is devoted to stating that multi-instance interpretability is the only desired interpretation one would like to have. This is false, if only for a very practical aspect: general data protection regulations require the right of explanation on the “single” prediction. The one that pertains the single individual and the single decision that has been automatically taken by a computational model. Than is single instance explainability and it is very important to develop methodologies and methods for it. POST-REBUTTAL: no comments or discussions on this in the rebuttal. I hope, nonetheless, that it will be taken into consideration in future version of the paper. - GNNExplainer is only capable of single-instance explainability: this is FALSE. There is not much to say but reference Appendix A of the original paper. I can agree that it is not natively thought to be multi-instance, but the paper conveys all the necessary information to use it in multi-instance settings. So please avoid excessive statements referring to an incapability of GNNExplainer in dealing with multi-instance. POST-REBUTTAL: Again here the issue is associated to an unclear positioning of the paper and of the related models in literature. In the rebuttal the authors clearly state that the difference between the proposed model and GNNExplainer is in the parameterization (which I agree on). But in the paper it is written that a key difference in that he proposed method is capable of multi-instance explanation while GNNEplainer is not. And this is not true, in absolute terms. So Authors are invited to revise this discussion. Presentation quality and level of technical detail is not adequate to allow the reader to reconstruct all the details of the model. This is due to the presentation lacking sufficient details when it comes to key aspect of the models and inconsistencies in the formalization. To gain sufficient insight into the model I have read throughout the appendixes: there I have found two pseudo-codes which did not help in clarifying open points. First, for the most part they are highly redundant, so reporting a single setting would have been sufficient, provided that the space saved is used to supply more information on the inner workings of the method. In particular, a second pseudo-code could have been used to show how to build the model level explanation. Second, equations (10) and (11) are defined on triples and couples of node embeddings, respectively. However, in their use in Algorithm 1 and 2 they are used within a loop running on a single node i: what about nodes j, v in (10) and j in (11)? All in all, there is not enough clarity in the description of the model to allow a its straightforward replication, and the lack of associated source code worsens this aspect. The empirical analysis has some unclear aspects: 1) Why is AUC used in place of accuracy as in the original GNNExplainer paper? Since the experimental setup is the same (as stated by the Authors) also the performance assessment should be the same (unless there is a very good reason for this, which I couldn’t find in the paper). 2) Why only PGExplainer results have stds? If all models have been tested under the same conditions, this should be available and reported for all in Table 1.

Correctness: Claims are seemingly correct. The empirical methodology seems mostly adequate, but requires clarification on two aspects as detailed in the weaknesses section above.

Clarity: Again, as detailed above, the paper has presentation issues which hinder understanding some key aspects of the model. The solution would be to rephrase and re-organise Section 4, paying particular attention to be consistent in the formalization of the model (within Section 4 and with respect to the Appendix). Also, the Appendix (although it is not subject of this review) could use a refactoring and rewriting to provide more details about the generation of the explanation subgraph. POST-REBUTTAL: I still find the pseudo-code unclear. And clearly I could not access the code on the github as I would have risked infringing anonymity.

Relation to Prior Work: The discussion of the relationship with works in literature is only partly satisfactory. As stated above, the paper should avoid stating that GNNExplainer cannot be used in multi-instance explanation (rather I would like to see a discussion on how this is not natural and comes at a cost). In addition, the paper is missing citation and comparison with some relevant recent works. In particular, there is one that has been published last year and should hence be included: https://openaccess.thecvf.com/content_CVPR_2019/papers/Pope_Explainability_Methods_for_Graph_Convolutional_Neural_Networks_CVPR_2019_paper.pdf The following work is highly related, but since it is from SIGKDD 2020 it is perfectly understood that the Authors have missed it. Nevertheless, it would be good to include it as well: https://arxiv.org/pdf/2006.02587.pdf POST-REBUTTAL: Even if the Authors believe that the works above cannot do everything that can be done with their approach, these still needs to be cited and discussed in the paper. Related literature needs citing and proper discussion.

Reproducibility: No

Additional Feedback: Apart from the issues described in previous fields, there is an additional aspect which requires some clarification. The Authors build their model on Gilber Random Graphs, which is based on a quite “strong” independence assumption. In the paper there is little discussion about this choice and about the potential consequences of this choice in the classes of explainable structures. For instance: is there a class of graphs in which you could expect the explainer to work best/worst? I am thinking about contrasting, for instance, molecular graphs with scale-free structures (e.g. social networks). A discussion on this aspect would help strengthen the paper. POST-REBUTTAL: no comment on this on the rebuttal. A linguistic revision is highly recommended, e.g.: - Qualitatively evaluation -> qualitative - Quantitatively evaluation -> quantitative


Review 2

Summary and Contributions: This problem builds on the GNNExplainer method for explaining graph neural networks (by identifying important subsets of the graph) in several important ways. It does so by using a parameterized model to produce the explanation for an arbitrary node/graph rather than resolving the optimization problem independently for each input. This gives both computational and performance improvements. # Update After Author Response The reviewer still thinks that this is a local explainer because "We provide an explanation for each instance" and because the authors follow the same evaluations scheme for GNNExplainer (a local explainer). Unless the single parameterized model that generates each of these local explanations is simple enough to be interpretable, this clearly makes the proposed method a method for generating local explanations (ie, a local explainer). When people look at each of these local explanations to gain a "general understanding/global perspective" of the model, they may or may not be able to tell if the explanations are generated "individually and independently" and the benefit of this (although probable to exist) is not tested in this work. One the surface, these are not major issues. But the field of interpretability is often murky enough and writing like this certainly contributes to that. The reviewer is uncertain about whether or not this is a serious enough issue to lower their score.

Strengths: This is an important problem and the proposed work seems to represent a significant quantitative improvement over existing methods. Overall, the presentation is easy to understand and the claims are justified.

Weaknesses: It's not clear that this is a global method and calling it this causes some confusion. While each explanation is given by the output of a single learned model rather solved for independently with its own optimization problem (which makes it more "global" in a sense), the explanations are still fundamentally local because they apply to a single node/graph. Some of consequences of this are: - Figure 1: It's unclear how the proposed method produces this type of explanation (which says "mutagens contain the NO2 group"). This seems like it requires "additional ad-hoc post-analysis ... to extract the shared motifs to explain a set of instances" [Line 48]. Perhaps this analysis is easier with the proposed method, but it still seems necessary. - Paragraph Starting in Line 185: This reads as if global explanations are strictly better than local explanations when they actually solve different problems. The reviewer believes this is trying to say something along the lines of "having a single (global) model that produces each explanation has advantages over solving an optimization problem independently for each explanation". - Line 236-238: It is unclear what metric is going to be used to compare local and global explanations. Generally, this comparison is challenging because local and global explanation solve different problems and (usually) do better at their respective problems and associated metrics. Given the metric described in the Paragraph Starting in Line 294, it seems like a local (per point) metric is used for both GNNExplainer and the proposed method.

Correctness: Overall, the claims seem to be empirically verified. I did not check the derivations in Section 4.2 carefully. "For the MUTAG dataset, we only consider the mutagen graphs because no explicit motifs exist in nonmutagenic ones." [Line 298] Does this mean the metric is measuring how often it identifies Carbon rings or NO2 groups to measure AUC? If so, it might be worth commenting on this. It seems like it probably is an incomplete (but reasonable) metric because there are other structures related to the task that aren't as ubiquitous.

Clarity: Overall, the paper is very well written and easy to read. More specific comments are in the next sections.

Relation to Prior Work: This work is very clear on how it is related to GNNExplainer, which seems to be the most closely related method. The grouping of explanation methods as either "white/black box" is a little unclear. "Whitebox mechanisms mainly focus on yielding explanations for individual predictions" [Line 87] and "Blackbox methods generate explanations by locally learning interpretable models" [Line 94]. These say relatively similar things, so it's the more detailed discussion afterwards that actually separate the methods.

Reproducibility: Yes

Additional Feedback: Equation 3: It might be worth clarifying that the "approximately equal" comes from the approximation of the distribution. How close it is to "equal" depends on how accurate the approximation is. Paragraph Starting at 185: It might be worth discussing the difference between "local fidelity" and "general understanding". As written, this reads as if global explanations are strictly better than local explanations when, in practice, they usually don't have the same local fidelity (the results in this work are surprising in this way). "On the other hand, some important edges are missing in the explanations given by GNNExplainer." [Line 288]. It'd be nice to discuss why this happens (although a smaller subset is sufficient for exactly that one point, a parameterized approach may "fill in" the missing pieces for more consistent explanations?).


Review 3

Summary and Contributions: The paper proposes a framework to explain the predictions made by Graph Neural Networks. In contrast to the state of the art methods such as GNNExplainer, the proposed method can be used in the inductive setting and it does not need re-training for every graph. Instead of learning to explain the prediction for each instance, the proposed method learns an explainer and shares it across instances. It can be viewed as learning a generator for explanations given graphs. It is an interesting approach with strong experimental results.

Strengths: 1. The approach looks quite promising and is supported by strong experimental results. - It shows consistent improvement against the best performance of baselines. The improvement is 4% ~ 24.6%. - The qualitative results are convincing as well. 2. By sharing the explanation network, the authors reduced the number of parameters. As a result, the explanation model size is independent to the number of edges or the size of graphs. 3. In addition to the smaller number of parameters, the inductive fashion leads to a drastic speed-up of the method since the proposed method does not need training for every instance. So the speed-up is 12x to 108x. It's huge. 4. The paper reads well and it is well-written.

Weaknesses: 1. It is understandable. Due to the limited space, the discussion about qualitative results is kind of short. The explanations by PGExplainer are more contiguous thanks to the connectivity constraints that are discussed in the supplement. If the main paper refers the supplement for the details, it will be more accessible for readers. 2. The local vs global argument are not fully demonstrated by experiments. The claim sounds strong at the beginning but the result section did not explicitly discuss about that. 3. If Table 1 has the GT graph, it will be easier to tell that the PGExplainer yields more accurate explanation than GNNExplainer. Motifs are kind of GTs but BA-Community and the cases with multiple motifs are not clear.

Correctness: Yes, the claims seem correct. It does not have any theoretical proof. The paper has intuition why their method works better and has a less number parameters than the baseline. All look good.

Clarity: Yes, it is well-written as mentioned in the strengths.

Relation to Prior Work: Yes, the authors successfully distance their work from the previous work especially GNNExplainer, which is a SOTA method. The strengths mentioned above have some of different aspects such as the shared model, generalizability in the inductive setting and so on.

Reproducibility: Yes

Additional Feedback: POST-REBUTTAL: Compared to GNNExplainer, the novelty of this paper may look insufficient. But the strong experimental results and speed-up by authors' parameteric formulation are noteworthy. As other reviewers pointed out, I believe that some of arguments should be toned down in the final version.


Review 4

Summary and Contributions: This work improves previous work "GNNExplainer" by adopting variational distribution while calculating MI between the subgraph and the original graph. The objective and criteria of the experiments remain the same as the previous work. The proposed "PGExplainer" learns a global parameter for producing statistics of variational distribution (binary concrete) and thus can be efficiently applied to new instances. # changes after author response I appreciate the authors' clarification on the contribution. Although I still think the "parameterized post-hoc explanation" itself is not novel enough for accepting the paper, but I decide to raise the score from 4 to 5 as I found the novelty in combining the various methods to make the parameterization acutally work and perform superior to the previous method. As other reviewers pointed out, I think the authors misused the concept of "multi-instance explanation". As GNNExplainer can perform a multi-instance explanation while not parameterizing the explanation itself, I guess "model-level explanation" is a more appropriate name for what the authors done in the work.

Strengths: The main contribution of this paper is to replace the original GNNExplainer's multivariate Bernoulli distribution with its relaxed version: binary concrete distribution, which is continuous relaxation of multivariate Bernoulli distribution. Thus the variational distribution can be updated end-to-end. As the distribution is learned with entire training examples, the explanation that PGExplainer creates is a lot sound than its single-instance competitor GNNExplainer.

Weaknesses: As GNNExplainer and PGExplainer are almost the same but the choice of subgraph sampling and its optimization, I found the significance of this paper is limited. The predecessor, GNNExplainer (1) suggested the MI formulation for GNN explanation, (2) created several synthetic datasets to evaluate its method. Compared to these contributions they made, I found PGExplainer's contribution is short.

Correctness: I found the claims and methods are correct.

Clarity: I found the paper is easy to follow.

Relation to Prior Work: Actually, GNNExplainer mentioned about their adaptation to multi-instance explanations through graph prototypes, which seems relevant to this work but omitted. Additionally, the total training time of PGExplainer is not reported, while the instance-wise training time of GNNExplainer is reported.

Reproducibility: Yes

Additional Feedback: typos : line 116 : $E$ -> $\mathcal{E}$ line 221 : Bsides -> Besides line 269 : (2) ATT utilizes -> (3) ATT utilizes

[Author Response · NeurIPS 2020]

We appreciate the valuable feedback from all the reviewers and will include the following discussion into our work.

**Q1. Discussion on GNNExplainer for multi-instance case (Reviewers #1 and #4).** As discussed in lines 47-52, to explain a set of instances, GNNExplainer first interprets a representative instance and then adopts ad-hoc post-analysis to generalize to multiple instances. We believe that this is not an elegant way to have a global view of the GNN model. "Since the explanatory motifs are not learned end-to-end, the model however may suffer from sub-optimal generalization performance." Instead, the explanation in PGExplainer is generated with a parameterized network, where parameters are shared among populations. PGExplainer is natively designed for collectively explaining multiple instances.

**Q2. Clarification for technical detail (Reviewer #1).** The inner workings of the method (how to sample the subgraph) are mainly described by equations. Note that index notations, such as $i$, in different contexts may have different meanings. Using graph classification as an example, in Algorithm 2, $i$ is the index of an instance (graph) to be explained. In line 8 of the algorithm, we iterate over all edges in the graph and each corresponding latent variable is calculated with Eq. (11). Since the index $(i, j)$ is used to indicate an edge in Eq. (11), we don't expand Eq. (11) to avoid confusion. The source code of PGExplainer can be found in GitHub with the name "PGExplainer".

**Q3. The usage of AUC (Reviewer #1).** We follow the experimental setting in GNNExplainer. In their paper, "explanation accuracy" is not formally defined. From their source code, we find that they actually used AUC (line 327, explainer/explain.py in their GitHub repository). Besides, comparing to classification accuracy, AUC is a more comprehensive measurement to evaluate binary classification.

**Q4. Std in Table 1 (Reviewer #1).** We didn't report std for baselines because they don't have sampling processes in subgraph generation. GNNExplainer doesn't report std in their paper either. Since PGExplainer includes sampling, std is reported to show its stableness. Baselines' stds are shown in the table below. (GRAD is deterministic with 0 std.)

**Q5. Discussion on related work (Reviewer #1).**

|  | BA-Shapes | BA-Community | Tree-Cycles | Tree-Grid | BA-2motifs | MUTAG |
|---|---|---|---|---|---|---|
| ATT | 0.815 ± 0.005 | 0.739 ± 0.015 | 0.824 ± 0.012 | 0.667 ± 0.005 | 0.674 ± 0.040 | 0.765 ± 0.013 |
| GNNExplainer | 0.925 ± 0.002 | 0.836 ± 0.001 | 0.948 ± 0.005 | 0.875 ± 0.012 | 0.742 ± 0.001 | 0.727 ± 0.014 |

Methods in the CVPR paper mentioned only explain a specific type of GNN: GCN, on a specific task: graph classification. PGExplainer is a general model compatible with different GNNs and diverse learning tasks. Besides, instead of edge-level important scores, they just calculate node-level important scores. Thus, this paper was not cited and compared by GNNExplainer. We select a method "Gradient" in the CVPR paper which doesn't require the global average pooling layer to calculate the importance of each node, then calculate the importance of an edge by average the connected nodes' importance scores. The AUC scores on BA-2motifs and MUTAG are 0.773 and 0.653, respectively, much lower than PGExplainer. The KDD paper mentioned just showed up (June 3). It only applies to graph classification. Second, it only provides model-level explanations without preserving the local fidelity. Thus, the generated explanation may not be a substructure of the real input graph. Instead, PGExplainer can provide an explanation for each instance with a global view of the GNN model, which can preserve the local fidelity.

**Q6. Usage of "global" (Reviewers #2 and #3).** We provide an explanation for each instance, which is generated from a parameterized network. Parameters are shared among instances in the population, which equips PGExplainer with a global view of the explained GNN model. That's why we call it a global method. For quantitative evaluation, we follow the setting in GNNExplainer to evaluate the explanation for each instance and then report the average AUC. In Fig. 1, we use a case study to show a high frequent motif in mutagens. For MUTAG dataset, we only evaluate with mutagens containing $NH_2$ or $NO_2$. In Table 1, "house" motif is used as GT in BA-Community (line 287).

**Q7. Discussion on "local fidelity" and "general understanding" (Reviewer #2).** Both "local fidelity" and "general understanding" are important to interpret a GNN model. As discussed in [38], "local fidelity" requires an explanation corresponds to how the model behaves in the vicinity of the predicted instance. For example, in MUTAG dataset, ubiquitous motifs for mutagens include $NH_2$ and $NO_2$ [50,9]. Some mutagens only have $NH_2$ and some only have $NO_2$. Thus, it is important to give an explanation for each instance. Meanwhile, "global perspective", or "general understanding" is important to ascertain trust in the mode [38]. However, by generating a painstakingly customized explanation for each instance **individually and independently**, it is challenging for GNNExplainer to have a "general understanding" of the GNN model, because the GNN model is trained with multiple instances. Thus, we proposed PGExplainer to preserve the "local fidelity", at the same time, with a global view of the GNN model. Lacking of "global" view of GNN model explains why some important edges are missing in the explanations given by GNNExplainer.

**Q8. Our contributions (Reviewer #4).** GNNExplainer is a pioneer to provide explanations for GNN's predictions. We include a parameterized network to enable explainer a global view of the GNN model. By collectively explaining a set of instances, PGExplainer can achieve more consistent and accurate explanations. Besides, since the parameterized network is shared among the population, PGExplainer is able to explain new instances without retraining. Empirically, PGExplainer is much more effective and efficient than the state-of-the-art method. It is non-trivial to significantly improve the accuracy performance with much less running time.

**Q9. Discussion on total training time of PGExplainer (Reviewer #4).** As discussed in Appendix D.1, "PGExplainer can achieve relatively good performance with a small number of trained instances, which makes PGExplainer more practical in large datasets. The results also explain why we dismiss the training time of PGExplainer and only count the inference time in Section 5" (line 541-543).

[Meta-Review · NeurIPS 2020]

The authors agree that the paper addresses an important problem and provides really strong empirical results. There was some discussion regarding the novelty of the approach; given that this is a crowded area, the reviewers strongly encourage the authors to clarify and appropriately place the proposed ideas amongst the related work. Additionally, the reviewers ask the authors to ensure reproducibility of the work, and avoid the characterization of their work as "global explanations".